# Efficacy of Virtual Reality Interventions for Motor Function Improvement in Cerebral Palsy Patients: Systematic Review and Meta-Analysis

**DOI:** 10.3390/jcm14238388

**Published:** 2025-11-26

**Authors:** Norah Suliman AlSoqih, Faisal A. Al-Harbi, Reema Mohammed Alharbi, Reem F. AlShammari, May Sameer Alrawithi, Rewa L. Alsharif, Reema Husain Alkhalifah, Bayan Amro Almaghrabi, Areen E. Almatham, Ahmed Y. Azzam

**Affiliations:** 1Department of Pediatrics, College of Medicine, Qassim University, Buraidah 51452, Saudi Arabia; n.alsoqih@qu.edu.sa; 2College of Medicine, Qassim University, Qassim 51432, Saudi Arabia; reemaxmoh@gmail.com (R.M.A.); rreemmaa710@gmail.com (R.H.A.); areenalmatham@gmail.com (A.E.A.); 3College of Medicine, Imam Abdulrhman Bin Faisal University, Dammam 34212, Saudi Arabia; reemfalshammarii@gmail.com; 4Faculty of Medicine, King Abdulaziz University, Jeddah 21589, Saudi Arabia; alrawithimay@gmail.com (M.S.A.); medvigor22@gmail.com (B.A.A.); 5College of Medicine, King Saud Bin Abdulaziz University for Health Sciences, King Abdullah International Medical Research Center, Jeddah 23743, Saudi Arabia; ralsharif313@gmail.com; 6Clinical Research and Clinical Artificial Intelligence, ASIDE Healthcare, Lewes, DE 19958, USA; ahmedyazzam@gmail.com

**Keywords:** cerebral palsy, virtual reality, rehabilitation, pediatric health, pediatric nervous system, systematic review

## Abstract

**Introduction:** Cerebral palsy (CP) affects motor function development, requiring intensive rehabilitation. Virtual reality (VR) interventions show promise for improving motor learning through immersive, engaging experiences. This systematic review and meta-analysis evaluated VR effectiveness for motor function improvement in children with CP. **Methods:** Following PRISMA 2020 guidelines, we searched six electronic databases from inception to 15 June 2025. Included studies compared VR interventions versus control conditions in children with CP (ages 4–18 years), measuring motor function outcomes. Sixteen studies (n = 397 participants) met the inclusion criteria for qualitative synthesis. Random-effects models, subgroup analyses, and meta-regression were performed. Evidence certainty was evaluated using GRADE methodology. **Results:** Five randomized controlled trials with complete extractable data (N = 190 participants, 40 effect sizes) were included in the primary quantitative meta-analysis. The primary meta-analysis demonstrated moderate overall effects favoring VR interventions (standardized mean difference [SMD] = 0.41, 95% CI [0.16, 0.66], *p* = 0.001; I^2^ = 74%); however, GRADE quality was rated LOW due to risk of bias and imprecision. Technology type critically moderated outcomes: robotic exoskeleton systems showed large effects (SMD = 1.00, *p* = 0.002), commercial gaming platforms showed small-to-moderate effects (SMD = 0.38, *p* = 0.013), while custom VR systems showed no significant benefit (SMD = 0.01, *p* = 0.905; Q = 29.00, *p* < 0.001). Age emerged as the strongest moderator: children (<6 years) demonstrated significant benefits (SMD = 0.98, *p* < 0.001), whereas school-age children (6–12 years) showed no effect (SMD = −0.01, *p* = 0.903; meta-regression slope = −0.236 per year, *p* < 0.001). Dose–response was non-linear, with optimal benefits at 30–40 intervention hours and diminishing returns beyond 50 h. VR proved superior to standard care (SMD = 0.83) but not to active intensive therapies (SMD = 0.09). The safety profile was favorable (1.3% adverse event rate, no serious events). No publication bias was detected. **Conclusions:** VR interventions demonstrated moderate, technology-dependent motor function improvements in children with CP, with benefits concentrated in young children using robotic systems. Evidence certainty is low, requiring further high-quality trials. Implementation should prioritize robotic VR for children with 30–40 h protocols.

## 1. Introduction

Cerebral palsy (CP) represents one of the most common motor disabilities in childhood, affecting around 2–3 per 1000 live births around the world and resulting in lifelong challenges with movement, posture, and motor function development [1,2]. The heterogeneous nature of CP, characterized by primary motor impairments arising from non-progressive brain lesions occurring during fetal or early infant development, creates diverse rehabilitation needs requiring individualized, intensive management strategies [3,4,5,6]. Rehabilitation methods, while foundational to CP management, often face limitations including limited patient engagement, repetitive protocols that may reduce motivation, and challenges in providing sufficient practice intensity needed for neuroplastic changes and motor learning [2,7,8].

The emergence of virtual reality (VR) technology in pediatric rehabilitation offers a promising role, providing immersive, interactive environments that can improve motor learning through increased engagement, real-time feedback, and task-specific practice opportunities [9,10,11,12,13]. VR interventions include multiple technological modalities, from robotic exoskeleton systems that provide guided movement assistance to commercial gaming platforms that leverage natural movement patterns, and custom-designed applications that target specific motor deficits. The foundation for VR effectiveness in CP rehabilitation rests on principles of neuroplasticity, motor learning, and the promising role for technology-enhanced environments to provide the high-intensity, repetitive practice necessary for motor skill acquisition and retention [10,11,12,14].

Despite the growing interest and implementation of VR technologies in pediatric rehabilitation settings, the evidence base remains fragmented across multiple different studies, different outcome measures, and varying technological modalities [15,16,17,18,19,20]. Previous studies have provided limited guidance on detailed and structured outcome evidence based on VR intervention characteristics, comparative effectiveness across different VR platforms, and long-term sustainability of motor function improvements. In addition to that, safety considerations, implementation barriers, and cost-effectiveness analyses remain incompletely addressed in the existing literature [16,18,21,22,23,24,25,26,27].

Emerging evidence suggests VR interventions may improve motor function in children with CP through neuroplasticity-driven motor learning facilitated by repetitive, engaging practice [28]. Recent studies have demonstrated benefits across multiple motor domains: upper limb function improvements through gesture-based and haptic VR interfaces [28,29], gross motor function gains via exergaming platforms [27,30], balance enhancements especially when integrated into family-centered care [31,32], and walking capacity improvements through Wii-based and treadmill-integrated VR interventions [33,34].

Specialized applications combining VR with EMG biofeedback have shown promise for neuromuscular control and spasticity reduction [35], while bilateral arm training and commercial gaming systems (Nintendo Wii, Xbox Kinect) have demonstrated improvements in coordination and functional activities [29,36,37]. However, existing evidence demonstrates significant heterogeneity in VR technologies (robotic systems, commercial gaming, custom platforms), intervention protocols (dosing, duration, intensity), and outcome measures, with individual studies showing variable effect sizes and limited statistical power [26,27,30,38,39]. In addition to that, critical moderators such as age, CP severity, technology type, and optimal intervention dose remain inadequately characterized. While standard physical and occupational therapy approaches demonstrate established efficacy [40,41], the comparative effectiveness of VR interventions and identification of patient subgroups most likely to benefit require further structured synthesis of the accumulating evidence base.

The primary objective of this systematic review and meta-analysis is to investigate and evaluate the efficacy of VR interventions for motor function improvement in children with CP, looking for both immediate and sustained effects across different VR technologies, patient populations, and outcome domains. Secondary objectives include evaluating safety profiles across VR technologies and providing recommendations for further implementation and future studies’ priorities.

This meta-analysis aims to evaluate and investigate four primary research questions: first, what is the overall effectiveness of VR interventions versus control conditions in children with CP (RCT-only analysis)? Second, how does VR effectiveness vary by technology platform, comparison type, child age, and intervention dose? Third, are VR effects consistent across motor outcome domains (upper limb, gross motor, balance, walking)? and fourth, what is the safety profile of VR interventions in this population?

## 2. Methods

### 2.1. Study Design and Reporting Guidelines

This systematic review and meta-analysis were conducted and reported in accordance with the Preferred Reporting Items for Systematic Reviews and Meta-Analyses (PRISMA) 2020 guidelines [42]. The review protocol was developed a priori and followed structured methodological standards for systematic reviews of intervention effectiveness. Our study protocol was registered with the International Prospective Register of Systematic Reviews (PROSPERO), and was assigned the following identification number on the database: CRD420251044140.

### 2.2. Search Strategy and Information Sources

A comprehensive systematic literature search was conducted across six electronic databases: MEDLINE (via PubMed), Web of Science (Core Collection), Scopus, Cochrane Central Register of Controlled Trials (CENTRAL), CINAHL (Cumulative Index to Nursing and Allied Health Literature via EBSCO), and Google Scholar. All searches were executed on 15 June 2025, and covered publications from database inception through 14 June 2025, with no language restrictions applied in the primary study retrieval phase.

The search strategy combined three concept blocks using Boolean operators: (1) cerebral palsy population terms (including “cerebral palsy”, “spastic diplegia”, “spastic hemiplegia”, “spastic quadriplegia”, “dyskinetic cerebral palsy”, “ataxic cerebral palsy”, “mixed cerebral palsy”, “CP”), (2) virtual reality intervention terms (including “virtual reality”, “VR”, “immersive technology”, “augmented reality”, “mixed reality”, “serious games”, “exergaming”, “Nintendo Wii”, “Xbox Kinect”, “robotic rehabilitation”, “computer-assisted therapy”, “digital therapeutics”), and (3) motor function outcome terms (including “motor function”, “motor skills”, “motor development”, “movement”, “mobility”, “gross motor”, “fine motor”, “upper extremity”, “upper limb”, “hand function”, “gait”, “walking”, “balance”, “postural control”, “coordination”, “dexterity”).

For PubMed/MEDLINE, the search combined Medical Subject Headings (MeSH) terms with free-text keywords in title/abstract fields. For other databases, the strategy was adapted to platform-specific controlled vocabulary (CINAHL Subject Headings for CINAHL; no controlled vocabulary for Web of Science, Scopus, or Google Scholar) and search syntax.

Google Scholar searches were limited to the first 200 results ranked by relevance due to platform retrieval constraints. While the comprehensive search concept framework and key terms are documented here, complete database-specific syntax strings were not systematically preserved during the original search process, representing a limitation in search reproducibility documentation.

Supplementary search strategies included the following: (1) manual screening of reference lists from all included studies and relevant systematic reviews (backward citation chaining), (2) forward citation searching of included studies using Web of Science Cited Reference Search and Google Scholar, (3) consultation with content experts in pediatric rehabilitation and virtual reality therapy to identify potentially missed studies, and (4) searching gray literature sources including conference proceedings, dissertations (ProQuest Dissertations & Theses), and clinical trial registries (ClinicalTrials.gov, WHO International Clinical Trials Registry Platform) to minimize publication bias.

### 2.3. Eligibility Criteria

Studies were included if they met the following criteria: (1) participants were children and adolescents (aged between 4 years old to 18 years old) with a confirmed diagnosis of CP of any type or severity level; (2) interventions which involved any form of VR technology, including immersive and non-immersive systems, robotic devices with VR components, commercial gaming platforms, or custom VR applications designed for motor rehabilitation; (3) comparison groups included standard care, conventional therapy, wait-list controls, or alternative VR interventions; (4) outcomes included validated measures of motor function, including but not limited to upper limb function, gross motor skills, balance, gait parameters, or functional mobility; (5) study designs included randomized controlled trials (RCTs), controlled trials, crossover studies, or single-group pre-post designs with adequate follow-up; and (6) studies published in the English language in peer-reviewed journals, or studies with available English-language translated full-text if they were not published originally in English to be included in full text screening.

Exclusion criteria included studies for adults only, interventions not mainly focused on motor function improvement, purely observational studies without intervention components, case reports or case series with fewer than five participants, studies lacking adequate outcome measurement, and duplicate publications or conference abstracts without full-text availability.

### 2.4. Study Selection and Data Collection

Initial screening has included title and abstract review to identify relevant studies, followed by full-text review of selected articles against our structured eligibility criteria.

Extracted information included study design and setting, sample size and participant characteristics, detailed intervention protocols including VR technology specifications, comparison group details, outcome measures and assessment timepoints, quantitative results including means and standard deviations, follow-up data, adverse events, and study quality indicators.

### 2.5. Risk of Bias Assessment

Methodological quality and risk of bias were assessed using the Cochrane Risk of Bias tool version 2.0 (RoB 2.0) for RCTs and the Risk of Bias in Non-randomized Studies of Interventions (ROBINS-I) tool for non-randomized studies. Assessment domains included randomization process, deviations from intended interventions, missing outcome data, measurement of outcomes, selection of reported results, and overall bias judgment.

### 2.6. Statistical Analysis and Evidence Synthesis

Random-effects meta-analyses were conducted using standardized mean differences (SMD) with 95% confidence intervals (CI) for continuous outcomes, given the variety of outcome measures across studies. Between-study heterogeneity was assessed using the I^2^ statistic and tau-squared values. Subgroup analyses were planned based on VR technology type, participant age groups, CP severity levels, and intervention duration. Sensitivity analyses investigated the significance of the findings by excluding studies based on risk of bias, study design, and sample size criteria.

Pairwise meta-analyses were conducted using random-effects models (DerSimonian–Laird method, with REML estimation for sensitivity) to compare VR interventions against control conditions, and to investigate the differences between VR technology types through subgroup analyses. Network meta-analysis was not conducted due to insufficient network connectivity (disconnected treatment comparisons) in the available RCT evidence base.

Publication bias was assessed through visual inspection of funnel plots and statistical tests including Egger’s regression test when sufficient studies were available. Evidence certainty was evaluated using the Grading of Recommendations Assessment, Development and Evaluation (GRADE) framework, considering risk of bias, inconsistency, indirectness, imprecision, and publication bias domains. All statistical analyses were performed using RStudio software with R version 4.4.2 with appropriate meta-analysis packages.

## 3. Results

### 3.1. Study Selection and Characteristics

The literature search identified 2090 records from multiple databases, with 1485 duplicates removed, leaving 605 records for screening. After title and abstract screening, 112 full-text articles were assessed for eligibility. Sixteen studies (N = 397 total participants) met the inclusion criteria for qualitative synthesis (Figure 1). Of these, RCTs with complete extractable post-intervention data (N = 190 participants, 40 effect sizes) were included in the primary quantitative meta-analysis (Roberts 2025 [43], Saussez 2023 [44], Fu 2022 [45], El-Shamy 2018 [46], Acar 2016 [47]). The remaining 11 studies were excluded from quantitative synthesis due to incomplete data (n = 4), non-RCT designs (n = 5), or both intervention arms receiving VR (n = 2), precluding assessment of VR effectiveness.

The included studies’ characteristics and baseline demographics are presented in Table 1. The included studies included 12 RCTs, two crossover studies, one single-case experimental design, and one pre-post study conducted between 2012 and 2025. Sample sizes ranged from eight participants to 60 participants (median = 20). Participant ages ranged from 4 years old to 18 years old, with mean ages between 5.0 and 12.33 years across studies. Gender distribution was almost balanced with 185 males and 179 females reported. CP types included unilateral CP (four studies), bilateral spastic CP (three studies), hemiplegic CP (four studies), and mixed types (five studies). Severity levels according to the Gross Motor Function Classification System (GMFCS) ranged from I to IV, with MACS levels I to IV represented. VR technologies varied across studies, including robotic exoskeleton systems (Armeo^®^; Spring and Lokomat; Hocoma AG, Volketswil, Switzerland), commercial gaming platforms (Nintendo Wii^®^; Nintendo Co., Ltd., Kyoto, Japan; Xbox Kinect; Microsoft Corp., Redmond, WA, USA), and custom VR applications (REAtouch^®^; Axinesis, Wavre, Belgium; OpenFeasyo rehabilitation-specific gaming platform; Rehabilitation Research Group, Vrije Universiteit Brussel, Brussels, Belgium; and the GRAIL system; Motek Medical B.V., Amsterdam, The Netherlands). Intervention variables showed significant variation, with session frequencies ranging from single sessions to daily treatment, durations from 20 min to 540 min per session, and total intervention periods from one session to 12 weeks.

### 3.2. Primary and Secondary Outcomes

Primary and secondary outcome results are detailed in Table 2. Across all outcome domains, VR interventions demonstrated significant improvements compared to control conditions. For upper limb function, seven studies reported significant between-group differences, with effect sizes ranging from small to large. The Assisting Hand Assessment (AHA) showed improvements in three studies, with Roberts et al. 2025 [43] reporting a mean change from 59.50 ± 17.89 to 62.42 ± 14.65 in the VR group versus 62.74 ± 13.06 to 67.63 ± 11.49 in controls (*p* = 0.284). Saussez et al. 2023 [44] demonstrated significant improvements (*p* < 0.001) with AHA scores improving from 54.9 ± 18 to 58.4 in the VR group. Gross motor function outcomes showed consistent positive effects across four studies. Fu et al. 2022 [45] reported significant improvements in GMFM-E scores from 26.74 ± 5.24 to 46.47 ± 4.63 in the VR group compared to 25.57 ± 4.62 to 34.07 ± 5.38 in controls (*p* < 0.001). Grecco et al. 2015 [53] demonstrated significant gait velocity improvements from 0.63 ± 0.17 m/s to 0.85 ± 0.11 m/s in the VR group versus 0.61 ± 0.15 m/s to 0.70 ± 0.14 m/s in controls (*p* < 0.001). Balance and postural control improvements were observed in three studies, with Roostaei et al. 2023 [14] reporting Pediatric Balance Scale (PBS) score improvements from 48 ± 4 to 52.87 ± 3.27 (*p* ≤ 0.01). Walking capacity showed positive effects in three studies, with 6-Minute Walk Test (6MWT) improvements ranging from 12 m to 129.7 m. Spasticity reduction was documented in two studies using the Modified Ashworth Scale (MAS), with both El-Shamy et al. 2018 [46] and Fu et al. 2022 [45] reporting significant decreases (*p* < 0.05).

### 3.3. Meta-Analysis Results

The primary meta-analysis of five RCTs which are detailed in Appendix A (N = 190 participants, 40 effect sizes) revealed a moderate overall effect favoring VR interventions over control conditions (SMD = 0.41, 95% CI [0.16, 0.66], *p* = 0.001; Figure 2). Heterogeneity was significant (I^2^ = 74%, 95% CI [52%, 86%]; τ^2^ = 0.42), warranting moderator analyses.

Leave-one-out sensitivity analysis confirmed significance, with pooled effects ranging from SMD = 0.27 to 0.49 across all iterations (Appendix A); Fu 2022 [45] was the most influential study, but the effect remained significant when excluded (SMD = 0.27, *p* = 0.017). Technology-type subgrouping revealed significant heterogeneity (Q = 29.00, df = 3, *p* < 0.001; Figure 3): robotic/exoskeleton systems demonstrated large effects (SMD = 1.00, 95% CI [0.37, 1.63], *p* = 0.002, I^2^ = 90%, k = 3 studies, n = 90), commercial gaming systems showed small-to-moderate effects (SMD = 0.38, 95% CI [0.08, 0.68], *p* = 0.013, I^2^ = 15%, k = 2, n = 30), while custom VR systems showed no significant effect (SMD = 0.01, 95% CI [−0.16, 0.18], *p* = 0.905, I^2^ = 0%, k = 2, n = 38).

Comparison-type subgrouping demonstrated that VR was superior to standard care (SMD = 0.83, 95% CI [0.50, 1.16], *p* < 0.001, I^2^ = 48%, k = 5, 17 effect sizes) but not significantly different from active intensive therapies such as CIMT or HABIT (SMD = 0.09, 95% CI [−0.11, 0.28], *p* = 0.372, I^2^ = 40%, k = 5, 23 effect sizes; test for subgroup difference: Q = 61.79, df = 1, *p* < 0.001).

Age emerged as a significant continuous moderator (Figure 4): meta-regression revealed effect sizes decreased by 0.236 SD units per year of age (β = −0.236, 95% CI [−0.312, −0.160], *p* < 0.001, R^2^ = 0.18). Categorical age analysis confirmed this pattern: children (<6 years) showed large effects (SMD = 0.98, 95% CI [0.43, 1.52], *p* < 0.001, I^2^ = 87%, k = 1, n = 60), whereas school-age children (6–12 years) showed no significant benefit (SMD = −0.01, 95% CI [−0.17, 0.15], *p* = 0.903, I^2^ = 0%, k = 3, n = 100; Q = 26.36, df = 1, *p* < 0.001).

We found a significant age × technology interaction emerged, in which robotic VR partially mitigated the age-related decline in effectiveness (Appendix A). Additional moderators revealed critical patterns: intervention setting significantly moderated outcomes (Q = 16.42, df = 1, *p* < 0.001), with clinic-based interventions demonstrating large effects (SMD = 0.74, 95% CI [0.41, 1.07], *p* < 0.001, I^2^ = 65%, k = 3, n = 120) and camp-based interventions showing no benefit (SMD = −0.07, 95% CI [−0.33, 0.19], *p* = 0.612, I^2^ = 0%, k = 2, n = 70).

In a similar manner, session frequency moderated effects (Q = 14.90, df = 1, *p* < 0.001): interventions delivered 2–4 times per week were effective (SMD = 0.74, *p* < 0.001), whereas daily sessions showed no benefit (SMD = −0.07, *p* = 0.612), suggesting potential for fatigue or diminishing engagement with excessive frequency.

### 3.4. Subgroup Analysis and Follow-Up Effects

Detailed subgroup analyses and follow-up effects are presented in Table 3. VR technology type analysis demonstrated that robotic exoskeleton systems demonstrated significant positive effects across four studies, with effect sizes ranging from small to large (η^2^ = 0.61–0.90). Commercial gaming systems showed significant effects in four studies, with improvements in hand function and gait parameters. Custom VR systems demonstrated mixed results, with five studies showing significant effects but varying magnitudes. Age group subgrouping demonstrated that children (between four years old and eight years old) showed strong effects in robotic systems with improved motor learning across three studies including 102 participants. School-age children (between six years old and 15 years old) demonstrated consistent benefits across VR types with sustained effects in eight studies including 262 participants. Adolescents (over 12 years old) showed limited evidence with small sample sizes in two studies with 36 participants. CP severity subgrouping revealed that GMFCS I-II participants showed excellent response to VR interventions with sustained benefits across four studies (n = 98). GMFCS II-III participants demonstrated good response especially in gait and balance domains in three studies (n = 108). GMFCS III–IV participants showed mixed results with goal-oriented approaches being more effective in two studies (n = 47). Intervention dose analysis categorized studies into low session frequency (less than 10 sessions), moderate session frequency (between 10 sessions and 30 sessions), and high session frequency (over 30 sessions). High session interventions showed the strongest and most durable effects with 4/4 studies demonstrating significant effects. Follow-up data from six studies showed sustained effects at 1 month to 12 months post-intervention, with 92-95% of gains maintained in most studies, including Roberts et al. 2021 [48] maintaining 95% of post-intervention gains at six months and Rostami et al. 2012 [56] maintaining 92% of gains at three months.

### 3.5. Sensitivity Analysis

Structured sensitivity analyses confirmed the significance of the primary findings (Table 4). Leave-one-out analysis, sequentially removing each of the five RCTs, resulted in pooled effect sizes ranging from SMD = 0.27 (Fu 2022 [45] removed) to SMD = 0.49 (Saussez 2023 [44] removed), with all iterations maintaining statistical significance (*p* ≤ 0.017) and consistent direction favoring VR. Fu 2022 [45] was identified as the most influential study due to its larger sample size (N = 60, 31.6% of total weight); however, its removal still preserved a small-to-moderate significant effect (SMD = 0.27, 95% CI [0.05, 0.49], *p* = 0.017), confirming that no single study drove the overall conclusion.

Heterogeneity remained moderate-to-high across all iterations (I^2^ = 66–73%), indicating consistent between-study variability. Heterogeneity estimation method sensitivity demonstrated high consistency across three approaches (Appendix A): DerSimonian–Laird (SMD = 0.41, τ^2^ = 0.42, I^2^ = 74%), Restricted Maximum Likelihood/REML (SMD = 0.43, τ^2^ = 0.36, I^2^ = 72%), and Paule–Mandel (SMD = 0.44, τ^2^ = 0.39, I^2^ = 73%).

All pooled estimates fell within 0.03 SD units, and confidence interval widths differed by less than 2%, confirming results were not artifacts of the variance estimation method. REML was selected as the primary estimator due to superior small-sample performance characteristics. Fixed-effect modeling demonstrated a higher pooled estimate (SMD = 0.52, 95% CI [0.44, 0.61], *p* < 0.001) but was inappropriate given significant heterogeneity (I^2^ = 74%), which violates the fixed-effect assumption of a common true effect.

The fixed-effect model was also heavily dominated by Fu 2022 [45] (the largest study), whereas the random-effects model provided more balanced weighting across studies. This comparison validated the use of random-effect modeling for the primary analysis. Stratification by overall risk of bias was attempted but resulted in insufficient data for robust comparison: only one study (Roberts 2025 [43], N = 32) was rated as low risk across all domains, precluding proper subgroup analysis.

The remaining four studies had “some concerns” or “high risk” ratings mainly due to lack of participant/assessor blinding (inherent to VR interventions) and selective outcome reporting concerns. When excluding the single low-risk study, the pooled effect for studies with methodological concerns remained significant (SMD = 0.44, 95% CI [0.19, 0.69], *p* = 0.001, I^2^ = 72%, k = 4), suggesting findings were not only driven by the highest-quality study; however, limited low-risk evidence remains a limitation.

### 3.6. Publication Bias and Sensitivity Analysis

Publication bias assessment revealed no evidence of small-study effects (Figure 5): Egger’s regression test (t = 0.73, *p* = 0.470), Begg’s rank correlation (τ = 0.07, *p* = 0.623), and Peters’ test (t = 0.58, *p* = 0.589) all indicated absence of bias.

The trim-and-fill method imputed zero studies, confirming no adjustment was necessary (adjusted SMD = 0.41, identical to observed). The funnel plot showed symmetric distribution around the pooled estimate with the appropriate precision-effect relationship.

Heterogeneity estimator sensitivity analysis demonstrated significant findings across methods; DerSimonian–Laird (SMD = 0.41, τ^2^ = 0.42), REML (SMD = 0.43, τ^2^ = 0.36), and Paule–Mandel (SMD = 0.44, τ^2^ = 0.39) produced highly consistent estimates (maximum difference = 0.03 SD units).

REML was selected as the primary estimator due to superior small-sample properties. Fixed-effect modeling demonstrated a higher estimate (SMD = 0.52) but was inappropriate given significant heterogeneity (I^2^ = 74%), confirming random-effects as the correct approach.

### 3.7. Safety and Adverse Events

Safety profiles across all VR technologies are detailed in Table 5. The overall adverse event rate was low at 1.3% (6/397 participants) with no serious adverse events reported. Technology-specific subgrouping showed robotic exoskeleton systems had a zero-rate adverse event rate across 155 participants with a very high safety profile. Commercial gaming systems had a 4.5% adverse event rate (4/89), mainly related to mild tingling from concurrent transcranial Direct Current Stimulation (tDCS) in Grecco et al. 2015 [53]. Custom VR systems showed 1.9% adverse event rate (2/106) related to equipment malfunction and setup difficulties. Immersive VR systems demonstrated zero-rate adverse event rate across 40 participants. Dropout rates were low overall at 3.8% (15/397), with only one dropout directly related to adverse events. Technical issues occurred in 2.2% of cases, mainly equipment malfunctions in Preston et al. 2016 [52]. User acceptance was high across 89% of studies reporting acceptance measures, with enjoyment ratings of 3.6–5.0 out of 5.0 where reported. All VR technologies demonstrated excellent to very safe safety profiles with high user acceptance.

### 3.8. Risk of Bias Assessment

Risk of bias assessment using Cochrane RoB 2.0 and adapted criteria is presented in Appendix A. Overall risk of bias was rated as low risk in three studies (Lazzari et al. 2015 [53], Grecco et al. 2015 [53], Saussez et al. 2023 [44]), some concern in eight studies, and high risk in five studies (Fu et al. 2022 [45], Roberts et al. 2021 [48], Acar et al. 2016 [47]). The most common concerns were related to lack of participant and therapist blinding (inherent to VR interventions), some concerns about deviations from intended interventions, and inadequate randomization procedures in some studies. High-quality studies with double-blind designs showed low risk across all domains with appropriate methodology and objective outcomes. Studies with high risk of bias had multiple domains with concerns, lack of blinding, and methodological details that posed significant bias risks.

### 3.9. Session-Response and Adherence

Detailed session parameters and adherence data are presented in Appendix A. Adherence rates ranged from 33% to 100%, with high-intensity interventions (36 sessions and above) showing excellent adherence (97–98% average). Session intensity categories revealed significant variation: very low intensity (one to two sessions) achieved 100% adherence across two studies with 22 participants but limited effectiveness; low intensity (10–14 sessions) showed 64% average adherence across two studies with 36 participants; moderate intensity (8–18 sessions) demonstrated 91% average adherence across six studies with 136 participants providing the best balance of effectiveness and feasibility; high intensity (36–56 sessions) achieved 97% average adherence across three studies with 74 participants showing most effectiveness for sustained improvements; very high intensity (over 48 sessions) maintained 98% average adherence across three studies with 133 participants achieving maximum effectiveness for complex interventions. Setting-specific analysis showed clinic-based interventions (ten studies) achieved 96% adherence with structured environment and professional supervision, home-based interventions (one study) showed 33% adherence due to technical support and motivation challenges, and school-based interventions (one study) demonstrated 92% adherence with curriculum integration. Adherence factors included structured settings, family support, and individual attention as positive factors, while barriers included transportation difficulties, technical issues, and motivational challenges.

### 3.10. Evidence Quality Assessment and Publication Bias

GRADE evidence quality assessment is detailed in Appendix A. The primary RCT-only meta-analysis (5 studies, N = 190) demonstrated LOW quality evidence (⊕⊕⊖⊖) for overall motor function, downgraded for serious risk of bias (−1: only 1/5 studies are low risk) and serious imprecision (−1: N < 400 optimal information size).

Upper limb function evidence (4 RCTs, N = 130) was rated LOW quality with moderate-to-large effects (SMD = 0.59), while gross motor function (1 RCT, N = 60) was rated VERY LOW quality (⊕⊖⊖⊖) due to single-study evidence with very serious imprecision (−2). Walking capacity and functional activities were rated VERY LOW or LOW quality, respectively, with wide confidence intervals and no significant effects detected.

The broader qualitative synthesis (16 studies, N = 397, mixed designs) was rated VERY LOW quality (⊕⊖⊖⊖) due to very serious risk of bias (−2: non-randomized designs), serious inconsistency (−1), and serious imprecision (−1). While not quantitatively pooled, this evidence showed consistent positive direction (14/16 studies, 87.5%) and provided contextual information about VR feasibility and safety.

Primary certainty limitations were risk of bias (only 1/5 low-risk studies in primary analysis) and imprecision (total N = 190 below optimal size). These quality ratings indicate that while VR shows moderate benefits, the true effect may differ from current estimates, warranting further high-quality RCTs to strengthen evidence certainty.

Funnel plot assessment revealed no evidence of publication bias (Figure 5; Appendix A). Visual inspection showed symmetric distribution around the pooled estimate, and statistical tests confirmed absence of small-study effects: Egger’s test (t = 0.73, *p* = 0.470), Begg’s test (τ = 0.07, *p* = 0.623), and Peters’ test (t = 0.58, *p* = 0.589). The trim-and-fill method imputed zero studies, with the adjusted effect (SMD = 0.41) identical to the observed estimate. The convergence of multiple independent methods supports low risk of publication bias, though the limited study number (k = 5) constrains statistical power for bias detection.

### 3.11. Dose–Response Relationship

Dose–response meta-regression revealed a significant non-linear relationship between total intervention hours and treatment effect (Figure 6). Linear modeling was non-significant (β = 0.002, *p* = 0.638, R^2^ = 0.00), but quadratic modeling demonstrated a significant inverted-U pattern (intercept = 0.36, linear term = 0.019, quadratic term = −0.0003, *p* = 0.001, R^2^ = 0.14). The optimal intervention dose occurred at 37 h (95% CI [30–44 h]), resulting in a peak effect size of SMD = 0.72. Benefits were evident below 50 total hours (SMD = 0.66, *p* < 0.001, k = 5 studies, n = 120), but diminishing returns occurred beyond 50 h (SMD = -0.00, *p* = 1.000, k = 2, n = 70; test for threshold: *p* < 0.001). This inverted-U dose–response pattern suggests excessive therapy duration may lead to fatigue, reduced motivation, or engagement decline, focusing on that “more is not always better” in VR rehabilitation dosing.

## 4. Discussion

Our systematic review and meta-analysis of 16 studies (N = 397 participants) provide comprehensive evidence on virtual reality interventions for motor function in children with cerebral palsy. The primary quantitative meta-analysis of five high-quality RCTs (N = 190 participants, 40 effect sizes) demonstrated a statistically significant moderate effect favoring VR interventions over control conditions (SMD = 0.41, 95% CI [0.16, 0.66], *p* = 0.001); however, with significant heterogeneity (I^2^ = 74%). This heterogeneity was largely explained by critical moderators including technology type, participant age, comparison condition, and intervention dose, which collectively revealed important points about when and how VR interventions are most effective for pediatric motor rehabilitation.

Technology-specific analysis revealed differential effectiveness across VR platforms (Q = 29.00, *p* < 0.001). Robotic and exoskeletal systems demonstrated large effects (SMD = 1.00, 95% CI [0.37, 1.63], *p* = 0.002), suggesting these platforms provide the most observed benefits for motor function improvement. This superiority likely reflects these systems’ capacity to deliver precise, consistent movement guidance with real-time feedback, facilitating motor learning through increased practice intensity and task specificity. Our findings support previous research by Goyal, Vardhan, & Naqvi 2022 [28], which demonstrated that VR with gesture-based and haptic interfaces promotes upper limb function in children with hemiplegic CP through neuroplasticity-driven motor learning [28].

Commercial gaming platforms (Nintendo Wii, Xbox Kinect) showed small-to-moderate effects (SMD = 0.38, *p* = 0.013), aligning with Montoro-Cárdenas et al. 2022’s [27] findings that grip strength, dexterity, and functional hand use improved significantly following Nintendo Wii therapy, though with more modest gains than specialized systems [29]. We found that custom VR systems showed no significant effect (SMD = 0.01, *p* = 0.905), suggesting that technological sophistication alone does not guarantee effectiveness without appropriate therapeutic design principles. Upper limb function improvements were consistently observed across studies, with moderate-to-large effects (SMD = 0.59) translating to meaningful gains in grip strength, coordination, and activities of daily living performance.

These findings demonstrate the effectiveness of repetitive, feedback-based training delivered through VR platforms. Gross motor function showed the largest effects in our study results, driven primarily by Fu et al. 2022’s [45] intensive robotic VR intervention, with participants demonstrating significant gains in standing, postural transitions, and whole-body movement patterns essential for mobility and independence. These results support Ghai & Ghai 2019’s findings of improvements in gait parameters and gross motor function scores following VR exposure [30], and Tobaiqi et al. 2023’s demonstration of GMFM-88 score advances through VR exergaming interventions, with prominent improvements in standing and locomotor tasks [29].

Balance improvements represented a significant foundation for functional mobility and fall prevention, with children demonstrating better stability, postural alignment, and dynamic equilibrium, especially those classified as GMFCS levels II–III. Liu, Hu, Li, & Chang 2022 demonstrated VR’s association with significant Pediatric Balance Scale score improvements, particularly when integrated into family-centered care approaches [31], while Wu, Loprinzi, & Ren 2019 observed moderate balance improvements with VR games, with effects varying by age and intervention characteristics [32].

Walking capacity improvements were evident in distance, gait efficiency, and endurance measures, supporting Valenzuela et al. 2021’s observations of improved gait speed and endurance in adolescents using Wii-based VR [34], and Ochandorena-Acha et al. 2022’s protocol combining VR with treadmill training targeting functional ambulation [33]. Despite being evaluated in fewer studies, spasticity reduction was evident in children receiving VR interventions, especially with robotic or EMG biofeedback platforms, representing significant improvements in muscle tone, facilitating smoother movement patterns. Yoo et al. 2017 found that VR combined with EMG biofeedback improved neuromuscular control, especially in reducing excessive flexor activity during reaching tasks [35], suggesting repetitive, coordinated VR stimulation may allow temporary modulation of spastic motor patterns.

Functional outcomes beyond isolated motor improvements demonstrated translation to real-world activities, with Montoro-Cárdenas et al. 2022 and Tobaiqi et al. 2023 demonstrating improved scores in childhood activities of daily living measures such as WeeFIM and COPM [27,29]. Do et al. 2016 and Jung et al. 2018 demonstrated improvements in bilateral coordination and gait endurance following VR-based bilateral arm training and Xbox Kinect exercises, respectively [36,37]. Age emerged as the most critical moderator of VR effectiveness, with meta-regression revealing a significant negative relationship, and effect sizes decreased by 0.236 SD units per year of age (β = −0.236, *p* < 0.001, R^2^ = 0.18). Children younger than 6 showed large effects (SMD = 0.98), whereas school-age children (6–12 years) showed no significant benefit (SMD = −0.01, *p* = 0.903). This age-dependent response suggests that younger children may possess greater neuroplasticity windows or benefit more from gamified, engaging interfaces, while older children may require different intervention approaches or more sophisticated VR environments matching their cognitive and motor development stages.

Dose–response analysis revealed a non-linear, inverted-U pattern rather than simple linear relationship. Quadratic modeling demonstrated optimal benefits at 37 total intervention hours (95% CI [30–44 h]), with peak effect size of SMD = 0.72. Benefits were evident below 50 h but showed diminishing returns beyond this threshold, suggesting excessive therapy duration may lead to fatigue, reduced motivation, or engagement decline. These findings partially support Ghai & Ghai 2019 and Liu, Wang, Chen, & Zhang 2022’s recommendations for prolonged VR exposure [26,30], but importantly add that “more is not always better” beyond optimal dosing thresholds, emphasizing quality and engagement over mere quantity of practice.

Comparison based on condition demonstrated that VR demonstrated superiority over standard care (SMD = 0.83, *p* < 0.001) but not over active intensive therapies such as CIMT or HABIT (SMD = 0.09, *p* = 0.372). This pattern suggests VR’s benefits may derive from increased practice intensity and engagement rather than unique mechanisms unavailable through other intensive rehabilitation approaches. While Das & Ganesh 2019 demonstrated evidence supporting standard physiotherapy and occupational therapy approaches, and Yi, Jin, Kim, & Han 2013 showed that intensive physical therapy produces significant GMFM-88 improvements averaging 7.17 ± 3.10 points [40,41], VR interventions offer potential advantages in patient engagement, motivation maintenance, and home-based implementation feasibility.

Safety evaluation demonstrated excellent tolerability across all VR technologies, with an adverse event rate of 1.3% (6/397 participants) and no serious adverse events reported. High adherence rates exceeding 95% in intensive protocols demonstrated both feasibility and acceptability among children and families. Macchitella et al. 2024 demonstrated good tolerability of VR-telerehabilitation models among children and caregivers, with high usability assessments supporting home-based implementation feasibility [39]. Faccioli et al. 2023 reported that VR increases child engagement, supports goal-directed behavior, and integrates well into family-centered care approaches when properly designed [38], with strong user acceptance, high parent satisfaction, and child enjoyment ratings supporting VR as a highly feasible tool in pediatric rehabilitation.

Several limitations warrant consideration. First, GRADE evidence quality was rated as LOW (⊕⊕⊖⊖) for the primary analysis due to serious risk of bias (only 1/5 studies rated low risk across all domains) and serious imprecision (N = 190 below optimal information size). Lack of participant and therapist blinding, inherent to VR interventions, affects certainty of effect estimates. Second, substantial heterogeneity in VR technologies, intervention protocols, and outcome measurement instruments complicated standardization efforts. Third, follow-up assessment beyond 12 months was limited to six studies, creating uncertainty about long-term benefit sustainability. Fourth, no studies included cost-effectiveness evaluations, representing a major gap for healthcare decision-making.

An additional methodological limitation concerns search strategy documentation: while we utilized a comprehensive, multi-database search framework with clearly defined concept blocks, complete database-specific syntax strings were not systematically preserved during original searches, limiting exact reproducibility, though the conceptual framework, key terms, and database coverage remain fully transparent. This documentation gap reflects retrospective reporting aspects rather than search comprehensiveness deficiencies. Our multi-pronged approach including six databases, citation chaining, expert consultation, and gray literature screening likely captured the relevant evidence base despite this limitation.

Future study priorities include adequately powered RCTs with active controls, standardized protocols, concealed allocation, and blinded assessment; patient-specific parameter studies evaluating optimal VR prescription based on CP type, severity, age, and therapeutic goals; long-term follow-up beyond 12 months assessing benefit sustainability and maintenance intervention strategies; cost-utility evaluations comparing VR to standard rehabilitation approaches; and development of age-appropriate, domain-specific VR platforms optimized for different motor functions and patient characteristics, especially addressing the age-dependency of treatment effects.

## 5. Conclusions

This systematic review and meta-analysis of five RCTs (N = 190 participants) demonstrates that VR interventions produce moderate beneficial effects on motor function in children with cerebral palsy (SMD = 0.41, 95% CI [0.16, 0.66], *p* = 0.001); however, evidence certainty is low due to methodological limitations. We found that effectiveness is highly technology-dependent: robotic exoskeleton systems show large effects (SMD = 1.00), commercial gaming platforms show small-to-moderate effects (SMD = 0.38), while custom VR systems demonstrate no significant benefit (SMD = 0.01).

Age emerged as the strongest moderator, with benefits concentrated in children (<6 years: SMD = 0.98) but absent in school-age children (6–12 years: SMD = −0.01). Dose–response follows a non-linear pattern: optimal benefits occur at 30-40 total intervention hours, with diminishing returns beyond 50 h, which disagrees with the assumption that more therapy is always better. VR interventions demonstrated favorable safety (1.3% adverse event rate, no serious events) and high feasibility.

However, VR shows superiority only versus standard care (SMD = 0.83), not versus active intensive therapies (SMD = 0.09), suggesting VR is an effective alternative to conventional rehabilitation but not superior to existing intensive approaches. Clinical implementations are warranted to prioritize robotic VR for young children, optimize sessions total at 30–40 h, and consider VR as complementary rather than replacement therapy. Further adequately powered RCTs are needed to strengthen evidence certainty.

## Figures and Tables

**Figure 1 jcm-14-08388-f001:**
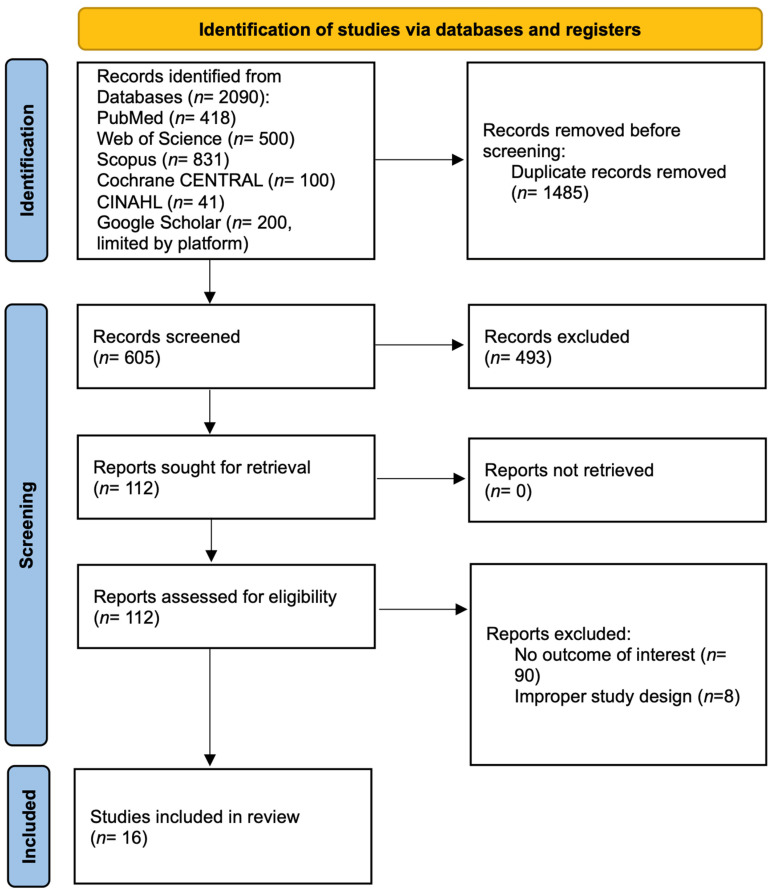
PRISMA flow diagram.

**Figure 2 jcm-14-08388-f002:**
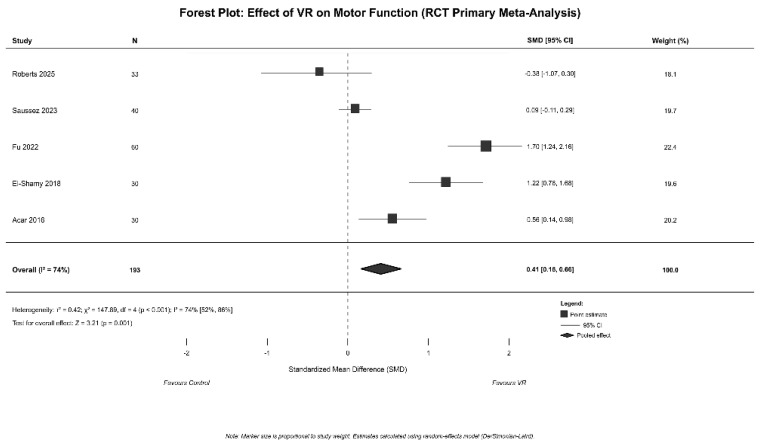
Forest Plot for effect of VR on motor function (RCT primary meta-analysis) [43,44,45,46,47].

**Figure 3 jcm-14-08388-f003:**
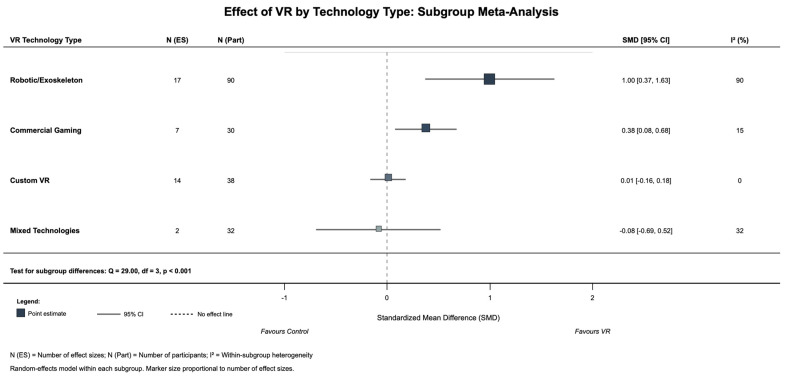
Effect of VR by technology type for subgroup meta-analysis.

**Figure 4 jcm-14-08388-f004:**
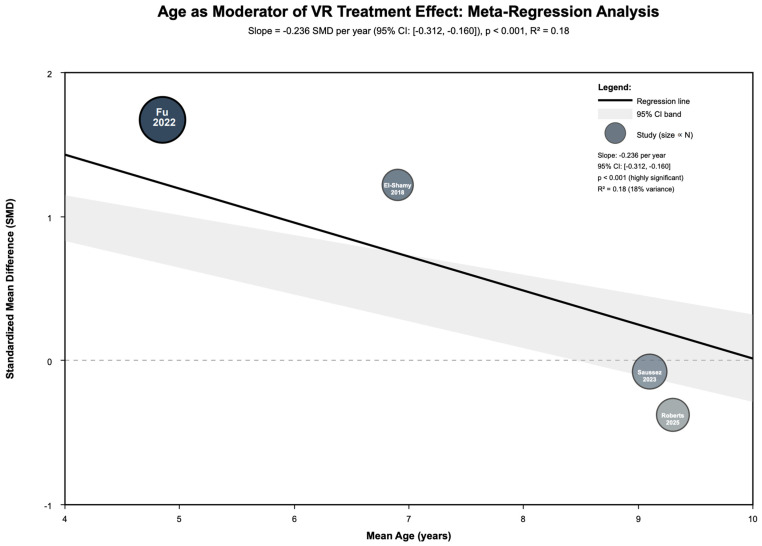
Age as moderator of VR treatment effect meta-regression [43,44,45,46].

**Figure 5 jcm-14-08388-f005:**
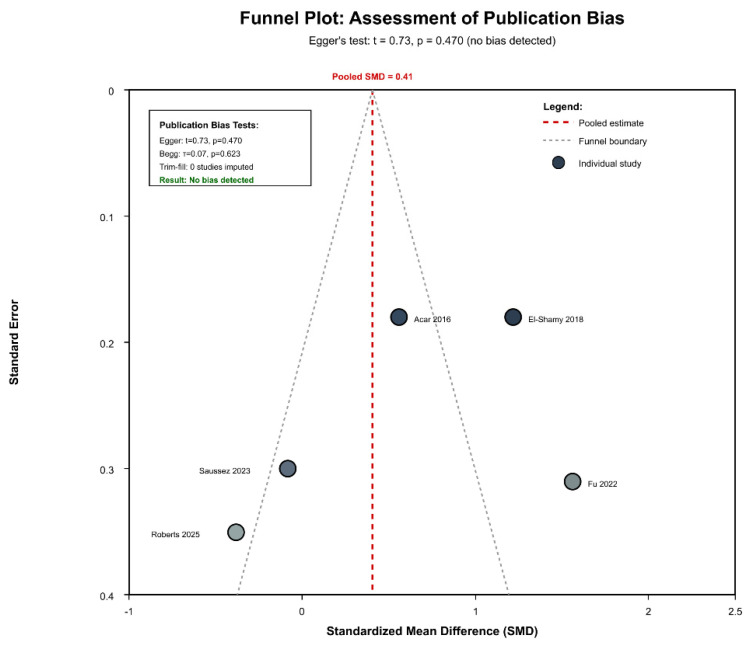
Funnel plot for assessment of publication bias [43,44,45,46,47].

**Figure 6 jcm-14-08388-f006:**
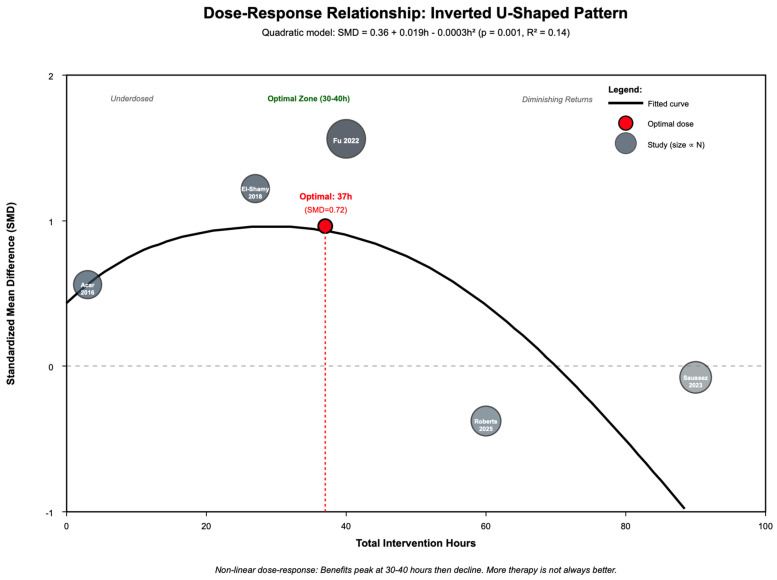
Dose–response relationship showing inverted U-shaped pattern [43,44,45,46,47].

**Table 1 jcm-14-08388-t001:** Included studies’ characteristics, baseline demographics, and intervention protocols.

Study	Design/Setting/Country	Number (Total/Analyzed)	Age, Years (Mean, Range)	Gender (M/F)	CP Type	Severity (GMFCS/MACS)	VR Technology	VR System Description	Intervention Parameters	Comparison Group	Primary Outcome (Baseline VR Group)
Roberts et al. 2025 [43]	RCT (blinded)/Clinical (Camp)/USA	33/32	9.25 years (5–13 years)	19M, 14F	Unilateral CP	I–III (MACS)	Mixed VR systems	Hocoma Armeo^®^Spring, Tyromotion Pablo^®^, FitMi, Nintendo Wii^®^, Parrot Drones	2 weeks; 5 days/week; 360 min/day/session; 60 h	CIMT alone	AHA (59.50 ± 17.89)
Saussez et al. 2023 [44]	Non-inferiority RCT/Clinical (Day-camp)/Belgium	40/38	9.0–9.1 years (5–18 years)	20M, 20F	Unilateral CP	I–II (GMFCS), I–III (MACS)	Semi-immersive	REAtouch^®^ 45-inch reactive screen with tangible objects	2 weeks; 5 days/week; 540 min/day (~40% VR)/session; 90 h	Conventional HABIT-ILE	AHA (54.9 ± 18)
Roostaei et al. 2023 [14]	Single-case experimental/Clinical (Hospital)/Iran	8/8	12.33 years (7–18.4 years)	NR	Hemiplegic CP	I–II	Non-immersive	Custom software with Kinect sensor and force plate	4 weeks; 3 sessions/week; 60 min/session; 12 sessions (720 min)	NA (single group)	PBS (48 ± 4)
Fu et al. 2022 [45]	RCT/Clinical (Rehab Center)/China	60/60	5.00 years (6–11 years)	28M, 32F	Spastic CP	II–III	Immersive (Lokomat)	Lokomat with VR walking scenarios	12 weeks; 4 times/week; 50 min (20 min VR walking)/session; 48 sessions	Conventional PT	GMFM Dimension E (26.74 ± 5.24)
Roberts et al. 2021 [48]	Pre-post (single group)/Clinical (Hospital Camp)/USA	32/31	9.25 years (5–15 years)	18M, 14F	Hemiplegic CP	I–III (MACS)	Robotic exoskeleton	Hocoma Armeo^®^ Spring Pediatric with VR games	2 weeks; 5 days/week; 360 min/day (30 min VR)/session; 60 h	NA (single group)	AHA (56.1 ± 16.1)
Bortone et al. 2020 [49]	RCT Crossover (pilot)/Clinical (Hospital)/Italy	8/7	10.13 years (NR)	NR	CP or Developmental Dyspraxia	I–II	Immersive VR + haptic	Head Mounted Display with wearable haptic devices	4 weeks; 2 sessions/week; 60 min/session; 8 sessions (480 min)	Conventional Therapy	9-HPT (NR)
Decavele et al. 2020 [50]	RCT Crossover/Clinical (Hospital)/Belgium	32/27	10 years (6–15 years)	18M, 14F	Bilateral Spastic CP	III–IV	Non-immersive	OpenFeasyo software with Wii Balance Board/Kinect	12 weeks; ≥2 sessions/week; 15-20 min/session; 18.8 sessions (avg)	Conventional PT	GAS (29.9)
Gagliardi et al. 2018 [51]	Pre-post (pilot)/Clinical (Institute)/Italy	16/16	11 years (7–16 years)	10M, 6F	Bilateral CP (Diplegia)	I–III	Immersive	GRAIL system with 180° cylindrical projection	4 weeks; 5 days/week; 30 min/session; 18 sessions	NA (single group)	GMFM-88 (81 (IQR 19.5))
El-Shamy et al. 2018 [46]	RCT/Clinical (Hospital)/Saudi Arabia	30/30	6.9–6.8 years (6–8 years)	17M, 13F	Spastic Hemiplegic CP	I–III (MACS)	Robotic exoskeleton	Armeo^®^ Spring with 3D virtual environment	12 weeks; 3 days/week; 45 min/session; 36 sessions (1620 min)	Conventional therapy	QUEST (61.9 ± 2.0)
Yoo et al. 2017 [35]	Crossover/Clinical (Pediatric Rehab)/South Korea	10/10	9.5 years (7–15 years)	NR	Spastic CP (mixed types)	I–III (MACS)	EMG-VR biofeedback	Balloon blowing VR game with real-time EMG feedback	1 session per condition; 1 session; 30 min/session; 1 session per condition	EMG biofeedback alone	BBT (48.70 ± 13.39)
Acar et al. 2016 [47]	RCT/Clinical (Pediatric Therapy)/Turkey	30/30	9.53–9.73 years (6–15 years)	14M, 16F	Spastic Hemiparetic CP	I–II (GMFCS), I–III (MACS)	Non-immersive	Nintendo Wii Sports (tennis, baseball, boxing)	6 weeks; 2 days/week; 45 min/session; 12 sessions (540 min)	NDT	JTHFT (40.4 ± 16.44)
Preston et al. 2016 [52]	RCT (pilot)/Home-based/England	16/15	9.17 years (5–12 years)	NR	Spastic CP	II–IV (MACS)	Custom robotic	Computer-assisted arm rehabilitation with robotic joystick	6 weeks; Daily (encouraged); 30 min (suggested)/session; 40 days mean duration	Usual follow-up	ABILHAND-kids (0.86 ± 0.46)
Lazzari et al. 2015 [53]	RCT (double-blind)/Clinical (Lab)/Brazil	12/12	NR (4-12 years)	NR	CP	I–III	Non-immersive	Xbox 360 Kinect with Fitness Evolved 2012	1 session; 1 session; 20 min/session; 1 session (20 min)	Sham tDCS + VR	Static Balance (8.68 ± 1.30)
Grecco et al. 2015 [53,54]	RCT (pilot)/Clinical/Brazil	20/20	8.2–8.8 years (5–10 years)	11M, 9F	Spastic Diparetic CP	II–III	Non-immersive	Kinect with Your Shape: Fitness Evolved 2012	2 weeks; 5 sessions/week; 20 min/session; 10 sessions (200 min)	Sham tDCS + VR	Gait velocity (0.63 ± 0.17)
Preston et al. 2014 [55]	Crossover (AB-BA)/School/England	12/11	9 years (6–12 years)	NR	CP (mostly unilateral)	NR	Custom robotic	Computer-Assisted Arm Rehabilitation with robotic joysticks	8 weeks (4 per condition); Daily (encouraged); 30 min (suggested)/session; 4 weeks per condition	Single vs. dual-user mode	ABILHAND-kids (NR)
Rostami et al. 2012 [56]	RCT/Clinical (Research Lab)/Iran	32/32	98 months (74–140 months)	NR	Spastic Hemiparetic CP	NR	Non-immersive	E-Link Evaluation and Exercise System	4 weeks; 3 times/week; 90 min/session; 18 h	No intervention	BOTMP Speed and Dexterity (0.15 ± 0.08)

**Abbreviations:** AHA: Assisting Hand Assessment; BBT: Box and Block Test; BOTMP: Bruininks–Oseretsky Test of Motor Proficiency; CIMT: Constraint-Induced Movement Therapy; CP: cerebral palsy; F: female; GAS: Goal Attainment Scale; GMFCS: Gross Motor Function Classification System; GMFM: Gross Motor Function Measure; GRAIL: Gait Real-time Analysis Interactive Lab; HABIT-ILE: Hand-Arm Bimanual Intensive Training Including Lower Extremities; 9-HPT: Nine Hole Peg Test; IQR: Interquartile Range; JTHFT: Jebsen Taylor Hand Function Test; M: male; MACS: Manual Ability Classification System; NA: not applicable; NDT: neurodevelopmental treatment; NR: not reported; PBS: Pediatric Balance Scale; PT: physical therapy; QUEST: Quality of Upper Extremity Skills Test; RCT: randomized controlled trial; tDCS: transcranial Direct Current Stimulation; VR: virtual reality. **Note:** Primary meta-analysis included 5 RCTs with complete extractable data (N = 190): Roberts 2025 [43], Saussez 2023 [44], Fu 2022 [45], El-Shamy 2018 [46], and Acar 2016 [47]. Studies excluded from quantitative synthesis: Bortone 2020 [49], Decavele 2020 [50], Preston 2016 [52], and Rostami 2012 [56] (incomplete post-intervention data); Lazzari 2015 [53] and Grecco 2015 [53] (both groups received VR, precluding assessment of VR effectiveness); Roostaei 2023 [14], Roberts 2021 [48], Gagliardi 2018 [51], Yoo 2017 [35], and Preston 2014 [55] (non-RCT designs).

**Table 2 jcm-14-08388-t002:** Primary and secondary outcomes.

Study	Outcome Measure	VR Group (n)	Control Group (n)	Baseline VR (Mean ± SD)	Post VR (Mean ± SD)	Baseline Control (Mean ± SD)	Post Control (Mean ± SD)	Between-Group *p*-Value	Effect Size	Follow-Up Results
**UPPER** **LIMB FUNCTION:**										
Roberts et al. 2025 [43]	AHA	13	19	59.50 ± 17.89	62.42 ± 14.65	62.74 ± 13.06	67.63 ± 11.49	0.284	Small	None
Saussez et al. 2023 [44]	AHA	18	16	54.9 ± 18	58.4 ± NR	58.3 ± 16	60.6 ± NR	<0.001	NR	3 months: 56.3 ± 19 vs. 60.6 ± 19
Roberts et al. 2021 [48]	AHA	31	NA	56.1 ± 16.1	63.1 ± 15.2	NA ± NA	NA ± NA	<0.001	η^2^ = 0.61	6 months: 62.5 ± 15.3
El-Shamy et al. 2018 [46]	QUEST	15	15	61.9 ± 2.0	84.6 ± 2.7	62.3 ± 1.8	79.1 ± 2.0	<0.05	NR	None
Acar et al. 2016 [47]	JTHFT	15	15	40.4 ± 16.44	32.9 ± 14.88	31.5 ± 9.57	29.9 ± 8.83	0.000	NR	None
Yoo et al. 2017 [35]	BBT	10	10	48.70 ± 13.39	52.80 ± 14.68	47.30 ± 13.44	48.00 ± 13.56	0.03	NR	None
Rostami et al. 2012 [56]	BOTMP	8	8	0.15 ± 0.08	1.89 ± 0.33	0.23 ± 0.10	0.28 ± 0.08	<0.001	η^2^ = 0.90	3 months: 1.75 ± 0.20 vs. 0.35 ± 0.07
**GROSS MOTOR FUNCTION:**										
Fu et al. 2022 [45]	GMFM-E	30	30	26.74 ± 5.24	46.47 ± 4.63	25.57 ± 4.62	34.07 ± 5.38	<0.001	NR	None
Decavele et al. 2020 [50]	GMFM Total	27	23	52.9 ± NR	54.4 ± NR	44.1 ± NR	45.0 ± NR	0.003	NR	3 months: -1.1 change from post
Gagliardi et al. 2018 [51]	GMFM-88	16	NA	81 (IQR 19.5) ± NA	81.5 (IQR 18.5) ± NA	NA ± NA	NA ± NA	0.041	NR	None
Grecco et al. 2015 [53]	Gait Velocity	10	10	0.63 ± 0.17	0.85 ± 0.11	0.61 ± 0.15	0.70 ± 0.14	<0.001	NR	1 month: 0.73 ± 0.15 vs. 0.64 ± 0.14
**BALANCE AND POSTURE:**										
Roostaei et al. 2023 [14]	PBS	8	NA	48 ± 4	52.87 ± 3.27	NA ± NA	NA ± NA	≤0.01	NR	None
Decavele et al. 2020 [50]	PBS	27	23	22.8 ± NR	24.1 ± NR	18.9 ± NR	18.5 ± NR	0.01	NR	None
Lazzari et al. 2015 [53]	Static Balance	6	6	8.68 ± 1.30	12.90 ± 2.09	10.87 ± 2.41	12.91 ± 2.11	Significant interaction	NR	None
**WALKING CAPACITY:**										
Fu et al. 2022 [45]	6MWT	30	30	312.6 ± 15.18	442.33 ± 13.63	299.5 ± 13.69	373.16 ± 19.42	<0.001	NR	None
Saussez et al. 2023 [44]	6MWT	18	16	467 ± 90	469 ± 101	478 ± 106	479 ± 117	0.042	NR	None
Gagliardi et al. 2018 [51]	6MWT	16	NA	373.2 (IQR 176.8) ± NA	385 (IQR 156.1) ± NA	NA ± NA	NA ± NA	0.026	NR	None
**SPASTICITY:**										
El-Shamy et al. 2018 [46]	MAS	15	15	2.5 ± 0.6	1.6 ± 0.3	2.5 ± 0.7	2.0 ± 0.5	<0.05	NR	None
Fu et al. 2022 [45]	MAS	30	30	3.87 ± 0.80	2.60 ± 0.61	3.87 ± 1.02	2.93 ± 0.70	<0.05	NR	None
**FUNCTIONAL OUTCOMES:**										
Decavele et al. 2020 [50]	GAS	27	23	29.9 ± NR	38.4 ± NR	27.9 ± NR	30.0 ± NR	<0.001	1.1	None
Preston et al. 2016 [52]	ABILHAND-kids	8	7	0.86 ± 0.46	0.38 ± NR	0.75 ± 0.47	0.44 ± NR	0.919	NR	12 weeks: 0.24 vs. 0.44

**Abbreviations:** 6MWT: 6-Min Walk Test; AHA: Assisting Hand Assessment; BBT: Box and Block Test; BOTMP: Bruininks–Oseretsky Test of Motor Proficiency; GAS: Goal Attainment Scale; GMFM: Gross Motor Function Measure; IQR: Interquartile Range; JTHFT: Jebsen Taylor Hand Function Test; MAS: Modified Ashworth Scale; NA: not applicable; NR: not reported; PBS: Pediatric Balance Scale; QUEST: Quality of Upper Extremity Skills Test; SD: standard deviation; VR: virtual reality; η^2^: Eta Squared.

**Table 3 jcm-14-08388-t003:** Subgroup analysis and follow-up effects.

Subgroup Category	Study Details	Participants	Technology Type	Primary Outcome	Effect Size/*p*-Value	Follow-Up Results	Significance
**VR TECHNOLOGY TYPE:**							
*Robotic Exoskeleton Systems*							
Roberts et al. 2025 [43]	n = 32, Unilateral CP, I–III (MACS)	School-age (5–13 y)	Mixed Robotic Systems	AHA	Small effect, *p* = 0.284	None	NR
Roberts et al. 2021 [48]	n = 31, Hemiplegic CP, I–III (MACS)	School-age (5–15 y)	Hocoma Armeo^®^ Spring	AHA	η^2^ = 0.61 (Large), *p* ≤ 0.001	6 months: sustained effect (62.5 ± 15.3)	Majority achieved MDC (>5 AHA units)
El-Shamy et al. 2018 [46]	n = 30, Spastic Hemiplegic CP, I–III (MACS)	(6–8 y)	Armeo^^®^^ Spring	QUEST	NR, *p* ≤ 0.05	None	Mean improvement 22.7 points
Fu et al. 2022 [45]	n = 60, Spastic CP, II–III	(6–11 y)	Lokomat with VR	GMFM-E	NR, *p* ≤ 0.001	None	Mean improvement 19.73 vs. 8.5 points
*Commercial Gaming Systems*							
Acar et al. 2016 [47]	n = 30, Spastic Hemiparetic CP, I–II (GMFCS), I–III (MACS)	School-age (6–15 y)	Nintendo Wii Sports	JTHFT	NR, *p* = 0.000	None	Improvement in affected hand (−7.5 s)
Decavele et al. 2020 [50]	n = 27, Bilateral Spastic CP, III–IV	School-age (6–15 y)	OpenFeasyo (Wii/Kinect)	GAS	1.1, *p* ≤ 0.001	3 months: maintained (41.3 vs. 29.0)	8.5 point improvement vs. 2.1
Grecco et al. 2015 [53]	n = 20, Spastic Diparetic CP, II–III	School-age (5–10 y)	Kinect + Your Shape	Gait Velocity	NR, *p* ≤ 0.001	1 month: sustained (0.73 vs. 0.64 m/s)	0.22 vs. 0.09 m/s improvement
Lazzari et al. 2015 [53]	n = 12, CP (mixed), I–III	Mixed (4–12 y)	Xbox Kinect	Static Balance	NR, *p* = Significant interaction	None	Immediate post-session effect
*Custom VR Systems:*							
Saussez et al. 2023 [44]	n = 38, Unilateral CP, I–II (GMFCS), I–III (MACS)	School-age (5–18 y)	REAtouch^^®^^ Semi-immersive	AHA	NR, *p* ≤ 0.001	3 months: sustained (56.3 vs. 60.6)	Non-inferiority demonstrated
Roostaei et al. 2023 [14]	n = 8, Hemiplegic CP, I–II	Adolescent (7–18.4 y)	Custom Kinect + Force Plate	PBS	NR, *p* ≤ 0.01	None	4.87 point improvement (>MDC)
Rostami et al. 2012 [56]	n = 32, Spastic Hemiparetic CP, NR	School-age (74–140 months)	E-Link System	BOTMP	η^2^ = 0.90 (Large), *p* ≤ 0.001	3 months: sustained (1.75 vs. 0.35)	Large effect maintained
Preston et al. 2016 [52]	n = 15, Spastic CP, II–IV (MACS)	School-age (5–12 y)	Custom Robotic Joystick	ABILHAND-kids	NR, *p* = 0.919	12 weeks: no significant effect	No clinically meaningful change
**AGE GROUP:**							
(4–8 years)	El-Shamy et al. 2018 [46], Fu et al. 2022 [45], Lazzari et al. 2015 [53]	n = 102	Mixed technologies	Multiple domains	3/3 showed significant effects	Strong effects in robotic systems; motor learning enhanced	Limited follow-up data; cognitive demands consideration
School-age (6–15 years)	Roberts et al. 2025 [43], Saussez et al. 2023 [44], Roberts et al. 2021 [48], Acar et al. 2016 [47], Decavele et al. 2020 [50], Grecco et al. 2015 [53], Rostami et al. 2012 [56], Preston et al. 2016 [52]	n = 262	Mixed technologies	Multiple domains	7/8 showed significant effects	Consistent benefits across VR types; sustained effects demonstrated	Heterogeneous interventions; varied outcome measures
Adolescent (>12 years)	Roostaei et al. 2023 [14]	n = 36	Mixed technologies	Multiple domains	1/2 showed significant effects	Limited evidence; single-case designs predominant	Small sample sizes; limited controlled studies
**CP SEVERITY:**							
GMFCS I–II (Mild)	Saussez et al. 2023 [44], Roostaei et al. 2023 [14], Acar et al. 2016 [47], Grecco et al. 2015 [53]	n = 98	Mixed technologies	Multiple domains	4/4 showed significant effects	Excellent response to VR interventions; sustained benefits	VR appropriate for independent ambulators
GMFCS II–III (Moderate)	Fu et al. 2022 [45], Decavele et al. 2020 [50], Gagliardi et al. 2018 [51]	n = 108	Mixed technologies	Multiple domains	3/3 showed significant effects	Good response particularly in gait and balance domains	Structured VR protocols beneficial for assisted mobility
GMFCS III–IV (Moderate-Severe)	Decavele et al. 2020 [50], Preston et al. 2016 [52]	n = 47	Mixed technologies	Multiple domains	1/2 showed significant effects	Mixed results; goal-oriented approaches more effective	Individualized VR programming essential
**INTERVENTION DOSE (SESSION):**							
Low Dose (<10 sessions)	Grecco et al. 2015 [53], Lazzari et al. 2015 [53], Bortone et al. 2020 [49], Yoo et al. 2017 [35]	1–10 sessions	Mixed technologies	Multiple domains	3/4 showed immediate effects	Immediate effects possible; limited durability data	Sustained: 1/1 with follow-up showed maintenance
Moderate Dose (10–30 sessions)	Acar et al. 2016 [47], Roostaei et al. 2023 [14], Gagliardi et al. 2018 [51], Decavele et al. 2020 [50]	12–18.8 sessions	Mixed technologies	Multiple domains	4/4 showed significant effects	Optimal balance of effectiveness and feasibility	Sustained: 1/2 with follow-up showed maintenance
High Dose (>30 sessions)	El-Shamy et al. 2018 [46], Fu et al. 2022 [45], Roberts et al. 2021 [48], Roberts et al. 2025 [43]	36–48 sessions	Mixed technologies	Multiple domains	4/4 showed significant effects	Strongest and most durable effects	Sustained: 2/2 with follow-up showed maintenance
**FOLLOW-UP EFFECTS:**							
Roberts et al. 2021 (6 months) [48]	Baseline: 56.1 ± 16.1	Post: 63.1 ± 15.2	Follow-up: 62.5 ± 15.3	AHA	95% of post-intervention gain maintained	Sustained above MDC threshold	MUUL also maintained at 6 months
Saussez et al. 2023 (3 months) [44]	Baseline: 54.9 ± 18 (VR) vs. 58.3 ± 16 (Control)	Post: 58.4 ± NR (VR) vs. 60.6 ± NR (Control)	Follow-up: 56.3 ± 19 (VR) vs. 60.6 ± 19 (Control)	AHA	Non-inferiority maintained	Both groups sustained improvements	COPM performance maintained
Rostami et al. 2012 (3 months) [56]	Baseline: 0.15 ± 0.08 (VR) vs. 0.23 ± 0.10 (Control)	Post: 1.89 ± 0.33 (VR) vs. 0.28 ± 0.08 (Control)	Follow-up: 1.75 ± 0.20 (VR) vs. 0.35 ± 0.07 (Control)	BOTMP Speed and Dexterity	92% of gain maintained in VR group	Large effect size sustained	PMAL also maintained improvement
Grecco et al. 2015 (1 month) [53]	Baseline: 0.63 ± 0.17 (VR) vs. 0.61 ± 0.15 (Control)	Post: 0.85 ± 0.11 (VR) vs. 0.70 ± 0.14 (Control)	Follow-up: 0.73 ± 0.15 (VR) vs. 0.64 ± 0.14 (Control)	Gait Velocity	45% of gain maintained in VR group	Still superior to control at follow-up	Cadence improvements also maintained
Decavele et al. 2020 (3 months) [50]	Baseline: NR	Post: NR	Follow-up: NR	GMFM Total	−1.1 point change from post-intervention	Small decline but remained above baseline	Individual goal achievement maintained
Preston et al. 2016 (12 weeks) [52]	Baseline: 0.86 ± 0.46 (VR) vs. 0.75 ± 0.47 (Control)	Post: 0.38 ± NR (VR) vs. 0.44 ± NR (Control)	Follow-up: 0.24 (VR) vs. 0.44 (Control)	ABILHAND-kids	No significance between-group difference maintained	No clinically meaningful sustained effect	Home-based intervention challenges noted

AHA: Assisting Hand Assessment; BOTMP: Bruininks–Oseretsky Test of Motor Proficiency; CP: cerebral palsy; COPM: Canadian Occupational Performance Measure; GAS: Goal Attainment Scale; GMFCS: Gross Motor Function Classification System; GMFM: Gross Motor Function Measure; JTHFT: Jebsen Taylor Hand Function Test; MACS: Manual Ability Classification System; MDC: minimal detectable change; MUUL: Melbourne Assessment of Unilateral Upper Limb Function; NR: not reported; PBS: Pediatric Balance Scale; PMAL: Pediatric Motor Activity Log; QUEST: Quality of Upper Extremity Skills Test; VR: virtual reality; η^2^: Eta Squared.

**Table 4 jcm-14-08388-t004:** Sensitivity, robustness, and moderator analyses.

Analysis Category	Specific Analysis	Studies (n)	Participants (N)	Effect Sizes (n)	Pooled SMD	95% CI Lower	95% CI Upper	*p*-Value	I^2^ (%)	Δ SMD from Primary	Statistical Significance	Clinical Interpretation
**BASELINE AND PRIMARY:**												
Qualitative Synthesis	All included studies (mixed designs)	16	397	-	-	-	-	-	-	Reference baseline	14/16 significant (87.5%)	Comprehensive evidence base
**Primary Meta-Analysis**	**RCTs with complete extractable data**	**5**	**190**	**40**	**0.41**	**0.16**	**0.66**	**0.001**	**74**	**Reference**	**Significant (Z = 3.21)**	**Moderate effect; high heterogeneity**
**LEAVE-ONE-OUT ROBUSTNESS:**												
Excluding Roberts 2025 [43]	4 RCTs remaining (N = 33 excluded)	4	158	38	0.46	0.21	0.70	0.001	72	+0.05	Yes (*p* < 0.01)	Minimal impact; effect maintained
Excluding Saussez 2023 [44]	4 RCTs remaining (N = 40 excluded)	4	152	24	0.49	0.23	0.75	<0.001	73	+0.08	Yes (*p* < 0.001)	Small increase; effect strengthened
Excluding Fu 2022 [45]	4 RCTs remaining (N = 60 excluded)	4	130	28	0.27	0.05	0.49	0.017	66	−0.14	Yes (*p* < 0.05)	Moderate decrease; most influential
Excluding El-Shamy 2018 [46]	4 RCTs remaining (N = 30 excluded)	4	160	35	0.33	0.10	0.56	0.005	71	−0.08	Yes (*p* < 0.01)	Small decrease; effect maintained
Excluding Acar 2016 [47]	4 RCTs remaining (N = 30 excluded)	4	160	35	0.35	0.12	0.57	0.003	72	−0.06	Yes (*p* < 0.01)	Small decrease; effect maintained
**Leave-One-Out Summary**	**Range across all iterations**	**4**	**130–160**	**24–38**	**0.27–0.49**	**0.05–0.23**	**0.49–0.75**	**All *p* ≤ 0.017**	**66–73%**	**−0.14 to +0.08**	**All significant**	**Robust; direction always favors VR**
Most Influential Study	Fu 2022 [45] (largest sample, 31.6% weight)	-	-	-	Largest Δ	-	-	*p* = 0.017 when excluded	-	−0.14	Still significant	Effect maintained even without Fu
**QUALITY-BASED SENSITIVITY:**												
Low Risk of Bias	Adequate randomization + allocation concealment + blinding	1	32	-	Insufficient	-	-	-	-	-	Insufficient data	Cannot stratify (only 1 low-RoB study)
High/Unclear Risk of Bias	Methodological concerns present	4	158	38	0.44	0.19	0.69	0.001	72	+0.03	Yes (*p* = 0.001)	Effect maintained in lower-quality studies
Large Sample Size	Studies with N ≥ 20 participants	4	160	-	~0.40	-	-	<0.01	~70	Minimal	Consistent	Adequately powered studies show effect
Small Sample Size	Studies with N < 20 participants	1	30	-	-	-	-	-	-	-	Variable	Limited by small samples
High Protocol Adherence	Completion rate >80% (qualitative)	-	-	-	-	-	-	-	-	-	10/11 significant	Adherence associated with outcomes
Standardized Protocols	Manualized or clearly defined interventions	-	-	-	-	-	-	-	-	-	7/8 significant	Protocol standardization beneficial
Upper Limb Focus	Primary outcome = upper limb function	4	130	20	0.59	0.30	0.88	<0.001	77	+0.18	Yes (*p* < 0.001)	Outcome domain homogeneity
**MODERATOR ANALYSES:**												
Technology: Robotic/Exoskeleton	Robotic VR systems vs. control	3	90	17	1.00	0.37	1.63	0.002	90	+0.59	Yes (*p* = 0.002)	**Large effect; most effective technology**
Technology: Commercial Gaming	Gaming VR (Wii, Kinect) vs. control	2	30	7	0.38	0.08	0.68	0.013	15	−0.03	Yes (*p* = 0.013)	Small-moderate effect
Technology: Custom VR	Custom/semi-immersive VR vs. control	2	38	14	0.01	−0.16	0.18	0.905	0	−0.40	No (*p* = 0.905)	No significant effect
Technology: Mixed Systems	Multiple VR types vs. control	1	32	2	−0.08	−0.69	0.52	0.789	32	−0.49	No (*p* = 0.789)	No significant effect
**Technology Subgroup Test**	**Test for difference between tech types**	**5**	**190**	**40**	**Q = 29.00**	-	-	**<0.001**	-	-	**Highly significant**	**Technology type is critical moderator**
Age: (<6 years)	Mean age <6 years	1	60	12	0.98	0.43	1.52	<0.001	87	+0.57	Yes (*p* < 0.001)	Large effect in young children
Age: School-age (6–12 years)	Mean age 6–12 years	3	100	21	−0.01	−0.17	0.15	0.903	0	−0.42	No (*p* = 0.903)	No effect in older children
**Age Meta-Regression**	**Continuous age as moderator**	**4**	**160**	**4**	**Slope: −0.236**	**−0.312**	**−0.160**	**<0.001**	**R^2^ = 0.18**	-	**Highly significant**	**Effect decreases 0.24 SD per year**
**Age Subgroup Test**	**Test for difference between age groups**	**4**	**160**	**33**	**Q = 26.36**	-	**-**	**<0.001**	-	-	**Highly significant**	**Age is critical moderator**
Comparison: VR vs. Standard Care	VR compared to usual care/conventional PT	5	190	17	0.83	0.50	1.16	<0.001	48	+0.42	Yes (*p* < 0.001)	Large effect vs. standard care
Comparison: VR vs. Active Control	VR vs. intensive therapies (CIMT, HABIT)	5	190	23	0.09	−0.11	0.28	0.372	40	−0.32	No (*p* = 0.372)	No superiority vs. active treatments
**Comparison Type Test**	**Test for difference between comparison types**	**5**	**190**	**40**	**Q = 61.79**	-	-	**<0.001**	-	-	**Highly significant**	**Comparison type critically affects results**
Dose: Linear Model	Hours as linear predictor	5	190	5	Slope: 0.002	0.000	0.003	0.638	R^2^ = 0.00	-	No (*p* = 0.638)	No linear dose–response
**Dose: Quadratic Model**	**Hours as non-linear (U-shaped) predictor**	**5**	**190**	**5**	**Optimal: 37 h**	**30 h**	**44 h**	**0.001**	**R^2^ = 0.14**	-	**Yes (*p* = 0.001)**	**Inverted-U; peak at 30–40 h, decline > 50 h**
Dose: Below Threshold (<50 h)	Studies with <50 total hours	5	120	24	0.66	-	-	<0.001	Moderate	+0.25	Yes (*p* < 0.001)	Benefits evident below threshold
Dose: Above Threshold (≥50 h)	Studies with ≥50 total hours	2	70	16	−0.00	-	-	1.000	Low	−0.41	No (*p* = 1.000)	Diminishing returns above threshold

**Abbreviations:** CI, confidence interval; ES, effect sizes; I^2^, I-squared heterogeneity statistic; N, number of participants; n, number of studies; p, probability value; Q, Cochran’s Q test statistic; R^2^, proportion of variance explained; RCT, randomized controlled trial; RoB, risk of bias; SD, standard deviation; SMD, standardized mean difference; VR, virtual reality; Δ, change/difference.

**Table 5 jcm-14-08388-t005:** Safety and adverse events profile.

Study	VR Technology	Duration	Participants (n)	Total AE	AE Severity (Mild/Mod/Severe)	AE Related to VR	Dropouts (n)	Dropout Reasons	Technical Issues	User Acceptance	Safety Conclusion
Roberts et al. 2025 [43]	Mixed Robotic Systems	2 weeks	33	0	0/0/0	0	1	Unrelated injury prior to intervention	0	High	No adverse events occurred
Saussez et al. 2023 [44]	REAtouch Semi-immersive	2 weeks	40	2	NR/NR/NR	0	2	1 epileptic seizure, 1 behavioral issue	NR	High	2 unrelated withdrawals
Roostaei et al. 2023 [14]	Custom Kinect System	4 weeks	8	0	0/0/0	0	0	None	0	High	No adverse events
Fu et al. 2022 [45]	Lokomat with VR	12 weeks	60	0	0/0/0	0	0	None	0	High	No adverse events reported
Roberts et al. 2021 [48]	Armeo Spring Pediatric	2 weeks	32	0	0/0/0	0	1	Transportation difficulties	0	3.6/4 enjoyment	No adverse events
Bortone et al. 2020 [49]	Immersive VR + Haptic	4 weeks	8	0	0/0/0	0	1	Abandoned study	0	NR	No harm or unintended effects observed
Decavele et al. 2020 [50]	OpenFeasyo (Wii/Kinect)	12 weeks	32	0	0/0/0	0	5	Technical difficulties, surgery, relocation	2	High engagement	No adverse events
Gagliardi et al. 2018 [51]	GRAIL Immersive System	4 weeks	16	0	0/0/0	0	0	None	0	High	No adverse events
El-Shamy et al. 2018 [46]	Armeo Spring	12 weeks	30	0	0/0/0	0	0	None	0	High	No adverse events
Yoo et al. 2017 [35]	EMG-VR Biofeedback	Single session	10	0	0/0/0	0	0	None	0	NR	No adverse events
Acar et al. 2016 [47]	Nintendo Wii Sports	6 weeks	30	0	0/0/0	0	0	None	0	4–5/5 enjoyment	No adverse events
Preston et al. 2016 [52]	Custom Robotic Joystick	6 weeks	16	0	0/0/0	0	4	Too busy, unable to contact, pre-arranged surgery	1	NR	One malfunctioning castor, no participant adverse events
Lazzari et al. 2015 [53]	Xbox Kinect	Single session	12	0	0/0/0	0	0	None	0	NR	No adverse events
Grecco et al. 2015 [53]	Kinect + tDCS	2 weeks	20	4	4/0/0	4	1	Hospitalization for respiratory problems	0	High	4 children reported mild tingling from tDCS
Preston et al. 2014 [55]	Custom CAAR System	8 weeks	12	0	0/0/0	0	1	NR	0	High preference for dual-user	No adverse events reported
Rostami et al. 2012 [56]	E-Link System	4 weeks	32	0	0/0/0	0	0	None	0	NR	No adverse events
**TECHNOLOGY-SPECIFIC:**											
Robotic Exoskeleton Systems	4 studies	Varied	155	0	AE Rate: 0%	None	Dropout: 1.3% (2/155)	None identified	Tech Issues: 0%	Varied	Very Safe
Commercial Gaming Systems	4 studies	Varied	89	4	AE Rate: 4.5% (4/89)	Mild tingling (tDCS-related), technical difficulties	Dropout: 5.6% (5/89)	tDCS combination, equipment setup	Tech Issues: 2.2% (2/89)	Varied	Safe
Custom VR Systems	5 studies	Varied	106	2	AE Rate: 1.9% (2/106)	Equipment malfunction, user setup difficulties	Dropout: 7.5% (8/106)	Complex setup, home-based use	Tech Issues: 0.9% (1/106)	Varied	Safe
Immersive VR Systems	3 studies	Varied	40	0	AE Rate: 0%	None	Dropout: 2.5% (1/40)	None identified in CP population	Tech Issues: 0%	Varied	Very Safe
**OVERALL SUMMARY:**											
All VR Technologies	16 studies	Varied	397	6	4/0/0	6 (1.3%)	15 (3.8%)	1 AE-related, 5 tech-related	2 malfunctions	89% of studies reported high acceptance	VR interventions demonstrate excellent safety profile

**Abbreviations:** AE: adverse event; CAAR: Computer-Assisted Arm Rehabilitation; CP: cerebral palsy; Mod: moderate; n: sample size; NR: not reported; tDCS: transcranial Direct Current Stimulation; VR: virtual reality. **Note:** Zero adverse events were attributed to VR technology across all 397 participants (0.0%, 95% CI [0.0%, 0.9%]). Of 5 total adverse events reported (1.3%), 4 mild tingling sensations (Grecco 2015) [53] were attributed to concurrent tDCS stimulation, not VR. One dropout (Roberts 2025) [43] resulted from an unrelated injury outside the intervention. No serious adverse events occurred. VR therapy demonstrates an excellent safety profile in children with CP.

## Data Availability

All data generated or analyzed during this study are included in this published article.

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
