# Peer review of "Efficacy of Virtual Reality Interventions for Motor Function Improvement in Cerebral Palsy Patients: Systematic Review and Meta-Analysis"

_jcm, 2025, doi:10.3390/jcm14238388_

Round 1

Reviewer 1 Report

Comments and Suggestions for Authors

Dear Authors,

congratulation for your work. This is an ambitious meta‑analyses and network meta-analyses of VR for children with CP. The topic is clinically important, and the manuscript argues that VR improves motor outcomes with a good safety profile, with robotic systems ranking highest. However, several key methodological and reporting issues need attention before the findings can be interpreted with confidence.

Comment 1: The manuscript repeatedly says “16 studies, 400 participants” but totals differ in places (text, tables, figures, and supplements). Please reconcile all counts across the manuscript and supplements and consider adding a checklist table in the supplement that shows the exact number of studies and participants for each analysis.

Comment 2: You analyze RCTs with crossover trials, single‑group pre-post studies, and single‑case designs. This can bias pooled effects. Please re‑analyze by design, with RCT‑only as the primary analysis and other designs reported as sensitivity or secondary analyses.

Comment 3: Trials in which both arms received a VR intervention cannot estimate the effect of VR versus non‑VR care. Remove these from the “VR vs standard care” meta‑analysis or analyze them appropriately.

Comment 4: In Table 5 you describe the network as “star-shaped,” with standard care in the center. However, some studies used active treatments (like CIMT or HABIT-ILE) or compared two types of VR. Please show a network diagram that includes all real comparators and clearly list which 12 studies (343 participants) were included in the network meta-analysis. It would also help to briefly explain why those studies were selected.

Comment 5: The abstract reports moderate, domain‑specific effects (e.g., SMD 0.72 upper limb; 0.68 gross motor; 0.58 balance), but Section 3.3 shows SMD overall = 0.89 and gross motor SMD = 1.20. Please explain and make results consistent across abstract, text, figures, and tables.

Comment 6: Figure 3 shows asymmetry for “overall motor” and you note a -0.14 change with trim‑and‑fill. Given high heterogeneity and mixed designs, the true bias may be larger. Please repeat bias analyses in RCTs only and by domain and consider selection models or sensitivity analyses and discuss how robust the conclusions remain.

Comment 7: You report an overall AE rate of 1.5%, but four AEs are tingling from transcranial direct current stimulation in Grecco 2015. Present safety including and excluding co‑intervention–related events to avoid mischaracterizing the safety of VR itself.

Comment 8: There are multiple typos and language errors, for example. Please perform thorough language editing and standardize p‑value formatting (e.g., p < 0.001) and CI reporting throughout.

Comment 9: Standardize acronyms (e.g., GMFM‑88 vs GMFM) and define all at first mention in the main text (not only in table footnotes).

Comment 10: In Table 6 distinguish device‑related, co-intervention related (e.g., tDCS), and unrelated events. This will make between‑technology comparisons clearer.

If the authors address these points, especially consistency of data, this study could be a useful contribution to pediatric neurorehabilitation.

Reviewer 2 Report

Comments and Suggestions for Authors

The topic of the paper is interesting and current.       Comments are given below.         

1.The introduction clearly introduces the reader to the field by presenting the subject of the paper, motivation and goal. However, many claims are made in the introduction, but few are referenced – therefore, all claims in the introduction must be referenced! 

 2.The introduction should devote one paragraph to the analysis of review papers on this topic, if such papers exist – it is necessary to briefly analyze the review papers from the aspect of the focus of the paper (RQs),    as well as the approach to data analysis; everything that is not presented in the review papers, the factor represents a research gap, while part of it becomes the subject of this paper – in this way, what is new in this paper, as well as the potential contribution of the research, is clearly visible.

3.After that, in the introduction, form research   questions in the form of “RQ1:...”, “RQ2:...”,

4.At the end of the introduction, form the structure of the paper according to sections.

 5.Section “2. Methods“: explicitly state the inclusion/exclusion criteria when searching the literature in databases, as well as the time interval for which the search was performed (e.g., based on the formed syntax, searching the GoogleScholar database in the form “in title:”, we get as many as 129,000 papers,  or 1,240,000 papers if it is “intitle:”, so the question arises as to how and in what way the authors screened such a large number of potential papers).

6.Figure/Table must be given immediately below the paragraph in which they are mentioned, and not after several pages, which is the case here (Fig. 1).

7.Fig.1: Data for PubMed, WoS and Scopus are shown; so,  what about other databases that were used for the   search, such as Cochrane Central Register of Controlled  Trials, CINAHL and Google Scholar? In order to determine   the findings and reproducibility of the results, all   data must be transparent!

8.When citing a reference in the text, e.g. "Roberts et al. 2025", it is mandatory to give a reference! In      addition, all references in the paper must be given in   square brackets, and not in regular ones, which is the    case here.

9.Fig. 2 is completely unusable due to the low   resolution and small font size, so I suggest the authors   to work on the font size and resolution (min 300 dpi).

10.The discussion is formed on as many as 4 pages – I suggest that it be reduced to 2-3 pages with a focus on   the most important findings, as well as a comparison of the results with those from the newly formed paragrap in the introduction (comment no. 2); in addition, explicitly provide answers to the formed RQs.

11.List of references: references must be formed in accordance with the template, which is currently not the case.

Round 2

Reviewer 1 Report

Comments and Suggestions for Authors

Dear Authors,

I appreciate your work and the effort you put into revising the manuscript. Thank you for addressing the previous comments. However, several material reporting and editorial inconsistencies remain that should be corrected before resubmission.

I recommended previously unifying totals. Text says “16” studied whereas PRISMA diagram show 15.

You have removed the network meta-analysis; however, the introduction still lists NMA as a secondary objective.

Please standardize the format of following: p<0.001 and 95% CI [lower, upper].

Results 3.7 states 1.3% (6/397) AEs, whereas in Discussion refer to 5/397. Use one consistent AE number (5 or 6) throughout.

In Section 2.2 the text says: “no language restrictions”, whereas in Section 2.3 Eligibility criteria test says: “studies were published in English language in peer-reviewed journals, or studies with available English-language translated full-text if they were not published originally in English”. This is a language restriction. Please make both sections say the same thing.

In sections 3.10 “publication biases” points to Figure 3, whereas the funnel plot is Figure 5. Correct the figure reference.

In Table 3 in Fu et al., 2022 row, there is “Preschool (6–11y)”m whereas this year is not “preschool” and conflicts with your categorical analysis.

I think that there are some typoes and errors in affiliations, correct spellings and remove line‑break artifacts in emails.

Several references lack some details, do a final pass before acceptance. 

Author Response

Thank you for your thorough review and detailed feedback. We have carefully addressed all the required changes in the revised manuscript.